DE-RALBA: dynamic enhanced resource aware load balancing algorithm for cloud computing

http://orcid.org/0000-0002-0558-1380 Hussain Altaf 1 altaf.hussain@ist.edu.pk
http://orcid.org/0000-0001-8342-5757 Aleem Muhammad 2
Ur Rehman Atiq 1
Arshad Umer 3
1 Department of Computer Science, KICSIT Campus, Institute of Space Technology , Islamabad , Pakistan
2 School of Computing, Mathematics and Data Science, Coventry University , London , United Kingdom
3 Faculty of Information Technology, Department of Computer Science, University of Central Punjab UCP , Lahore , Pakistan
Pasqua Michele
Electronic publication date: 2025 Mar 18
Publication date: 2025
Volume: 11
Electronic Location ID: e2739
Received 2024 Oct 3; Accepted 2025 Feb 10
Copyright: © 2025 Hussain et al.
Copyright year: 2025
Copyright holder: Hussain et al.
License: This is an open access article distributed under the terms of the Creative Commons Attribution License, which permits unrestricted use, distribution, reproduction and adaptation in any medium and for any purpose provided that it is properly attributed. For attribution, the original author(s), title, publication source (PeerJ Computer Science) and either DOI or URL of the article must be cited.
License URL: https://creativecommons.org/licenses/by/4.0/

Keywords: Cloud computing, Load balancing, Scheduling algorithm, Resource-aware scheduling, Resource-aware load balancing, Dynamic scheduling, Dynamic load balancing, Distributed computing, DE-RALBA

Funding: The authors received no funding for this work.

==============================
Cloud computing provides an opportunity to gain access to the large-scale and high-speed resources without establishing your own computing infrastructure for executing the high-performance computing (HPC) applications. Cloud has the computing resources (i.e., computation power, storage, operating system, network, and database etc.) as a public utility and provides services to the end users on a pay-as-you-go model. From past several years, the efficient utilization of resources on a compute cloud has become a prime interest for the scientific community. One of the key reasons behind inefficient resource utilization is the imbalance distribution of workload while executing the HPC applications in a heterogenous computing environment. The static scheduling technique usually produces lower resource utilization and higher makespan, while the dynamic scheduling achieves better resource utilization and load-balancing by incorporating a dynamic resource pool. The dynamic techniques lead to increased overhead by requiring a continuous system monitoring, job requirement assessments and real-time allocation decisions. This additional load has the potential to impact the performance and responsiveness on computing system. In this article, a dynamic enhanced resource-aware load balancing algorithm (DE-RALBA) is proposed to mitigate the load-imbalance in job scheduling by considering the computing capabilities of all VMs in cloud computing. The empirical assessments are performed on CloudSim simulator using instances of two scientific benchmark datasets (i.e., heterogeneous computing scheduling problems (HCSP) instances and Google Cloud Jobs (GoCJ) dataset). The obtained results revealed that the DE-RALBA mitigates the load imbalance and provides a significant improvement in terms of makespan and resource utilization against existing algorithms, namely PSSLB, PSSELB, Dynamic MaxMin, and DRALBA. Using HCSP instances, the DE-RALBA algorithm achieves up to 52.35% improved resources utilization as compared to existing technique, while more superior resource utilization is achieved using the GoCJ dataset.

Introduction

Cloud computing (CC) is a business-oriented model for providing services as a utility (Silva Filho et al., 2017). Services such as computation power, storage, operating system, network, and databases are provided on demand like a public utility over the Internet (Manvi & Shyam, 2014; Calheiros et al., 2011) to cloud users on pay-as-you-go policy. Cloud computing refers to networked computers comprising more than one unified computing resource (Arunarani, Manjula & Sugumaran, 2019). In recent years, the development of cloud computing has assisted in replicating the rapid installation of globally scattered, interconnected data-centers that provide high-quality and dependable services (Arshad et al., 2022). With its expanding use and marketing, cloud computing offers enormous opportunities and poses significant challenges to the development of traditional IT. Cloud service providers (CSPs) like Amazon, Microsoft, and Google, etc. supply their clients with cloud computing services and facilities to ensure quality services according to a specific business model. A web browser allows customers to reach online services in various domains, including business, education, and government. At the same time, data and software applications remain on cloud servers in data-centers (Rashid & Chaturvedi, 2019) and provide scalable and elastic resources. Cloud resources are provisioned in a virtualized form on the physical host as per the requirement of end-users (Masdari, Nabavi & Ahmadi, 2016). Virtualization hides the heterogeneity of the combined resources.

Cloud scheduling techniques are challenging and important for gaining good performance in high-Performance computing (HPC) (Hussain et al., 2018). In cloud computing, scheduling techniques can be categorized into static or dynamic approaches. In static technique, the order and timing of the jobs are determined at the start prior to the execution begins. Static scheduling has less run-time overhead than dynamic techniques, which have less or no information about the arrival of the jobs at the start. In dynamic techniques, the allocation of jobs completes at the run-time by calculating the execution time upon arrival (Ibrahim, El-Bahnasawy & Omara, 2017). The Min-Min heuristic favors smaller jobs and maintains a good makespan for smaller ones; while the makespan will be longer for the important large-sized jobs in the job pool (Arshed & Ahmed, 2021; Son, He & Buyya, 2019). The Max-Min heuristic aims to allocate large-size jobs in the workload among resources first but this may result in delayed allocation of smaller jobs to the resources (penalizing smaller jobs in the pool) (Hussain et al., 2018). The dynamic load balancing algorithm minimizes the makespan and ensures scalability when the datacenter’s load exceeds capacity. A dynamic load balancing algorithm (Kumar & Sharma, 2020) optimizes the effective utilization of cloud resources to enhance application execution speed, reducing makespan, and introducing the elasticity to the cloud environment. However, this approach needs to improve the potential overhead (consumes in continuously monitoring and reallocating resources) that could impact overall system performance and responsiveness. Additionally, it requires help to adapt quickly to sudden changes in the workload, potentially leading to sub-optimal resource allocation. A load balancing algorithm (Shafiq et al., 2021) involves intelligent workload placement strategies considering the applications’ nature and resource demands, striving to achieve better load distribution and improved cloud capacity utilization. Machine learning-based techniques can also predict resource demands and optimize job placement for enhanced load balancing and performance in cloud environments. A potential weakness lies in the complexity of these strategies, as they might require significant computational resources and intricate decision-making processes, which could reduce the algorithm’s efficiency and ease of implementation. The shortest round vibrant queue (SRVQ) algorithm (Siddesha & Jayaramaiah, 2021) is a fusion of the round-robin and shortest-job-first algorithms within the context of the vibrant quantum methodology. SRVQ aims to minimize waiting times during the scheduling process and alleviate issues related to task starvation. In the final experimental results, the collaborative use of dynamic voltage and frequency scaling (DVFS) with SRVQ demonstrated optimizes the effective utilization of cloud resources. The complications of combining two scheduling algorithms might introduce complexity into its implementation and maintenance, which could impact its efficiency in real-world dynamic cloud environments.

This article proposes a dynamic enhanced resource-aware load balancing algorithm (DE-RALBA) to reduce the load imbalance in the cloud environment as a modified version of RALBA (Hussain et al., 2018). The DE-RALBA mitigates the load imbalance, minimizes the makespan, and maximizes the throughput by maintaining a VM and Job-status tables to allocate the job to resources at run-time more efficiently. DE-RALBA gives the job to a virtual machine (VM) based on the computation powers of VMs and earlier finish time (EFT).

The contributions of this article are described as follows: A comprehensive literature review to explore the key features of the dynamic resource scheduling algorithm in literature;

Identifying a need for load-balancing based on resource-aware scheduling that leads to propose a dynamic approach named DE-RALBA reducing load imbalance by providing improved resource utilization and makespan in the cloud computing;

Providing empirical assessment of the proposed scheme DE-RALBA compared to state-of-the-art scheduling algorithms i.e., PSSLB (Alaei & Safi-Esfahani, 2018), PSSELB (Alaei & Safi-Esfahani, 2018), Dynamic MaxMin (Mao, Chen & Li, 2014) and DRALBA (Nabi, Ibrahim & Jimenez, 2021).

The article is structured as follows; the section 2 explains the related work. The proposed DE-RALBA presents the model, system architecture and DE-RALBA algorithm in detail. The experimental setup, workload characteristics, attained results and performance evaluation are discussed in the experimental evaluation section. The last section explains the conclusion and future work of the article.

Related work

This section discusses and critically presents the working semantics of the prominent state-of-the-art cloud scheduling algorithms as following.

Min-Min is a scheduling algorithm that depends on minimum completion time (MCT) (Elzeki, Rashad & Elsoud, 2012). The job is assigned to a resource having a minimum expected completion time. The scheduler maintains a matrix on pending jobs and calculates the ECT of all jobs against all the resources. Sequentially, jobs are assigned to a resource based on MCT and removed from the pending list. The algorithm does not update the status of the VM at run-time and assigns a maximum number of jobs to a resource solely based on its maximum computation power; it can indeed lead to an imbalanced load distribution (Hussain et al., 2018). Comparatively, Max-Min performs the job scheduling similarly while the main difference is in the job selection that is based on the maximum completion time (Hussain et al., 2018; Tabak, Cambazoglu & Aykanat, 2013). Max-Min algorithm maintains a matrix on pending jobs and calculates all jobs’ ECT against all the resources. First, Max-Min picks the job with maximum completion time and then select the resource that provides an earlier finish time on the selected job (Laroui et al., 2021). This process continues until all jobs are scheduled to the available resources. For the workload comprising smaller jobs, Max-Min may produce imbalance by delaying the scheduling of smaller jobs (Hussain et al., 2018; Tabak, Cambazoglu & Aykanat, 2013). Dynamic Max-Min is an extended version of Max-Min (Mao, Chen & Li, 2014) and makes cloud computing more flexible. It chooses the job that will take the longest to complete and allocates it to the resources that give the earliest finish time. In the dynamic Max-Min approach, the VM status table is used to analyze the workload of each VM and the completion time of jobs. This analysis is performed at runtime, and the status table is updated periodically at regular intervals. The dynamic Max-Min increases resource utilization and decreases response time for a jobs (Nabi, Ibrahim & Jimenez, 2021). An advanced Max-Min algorithm for task scheduling is introduced for cloud computing, aiming to optimize makespan, waiting time, and resource utilization (Raeisi-Varzaneh et al., 2024). By building on traditional Max–Min methods, the proposed approach balances execution efficiency and resource usage while integrating a cost-aware component to manage task execution costs effectively. The algorithm dynamically adjusts task allocation to meet user requirements within cloud provider constraints, and enhancing overall economic efficiency. Results are superior to traditional Max–Min, Min–Min, and SJF algorithms.

Proactive simulation-based scheduling and enhanced load balancing (PSSELB) and proactive simulation-based scheduling and load balancing (PSSLB) (Alaei & Safi-Esfahani, 2018) are scheduling algorithms that build upon Max-Min heuristics. PSSLB calculates the ECT of jobs against all the resources and selects the larger jobs to assign to the resource providing the lesser execution time. The process is iterated until all jobs are allocated to the computing resources. PSSELB is an extended version of PSSLB and offers more load balance. PSSELB uses the PSSLB approach to assign the jobs to the resources with the main difference of pivot element. PSSELB selects the larger-sized job and transfers it to the resource with EFT; a job’s completion time is considered as a pivot. In the subsequent iterations, the pivot is used to select jobs with equal and minimum completion times on other resources. The next larger is selected by PSSELB using PSSLB until no more jobs are left. PSSLB and PSSELB give high resource utilization and better makespan and resource utilization (Ibrahim et al., 2020). PSSLB and PSSELB produce a poor makespan on the GoCJ dataset (with higher number of jobs), limiting their versatility.

The resource aware load-balancing algorithm (RALBA) (Hussain et al., 2018) assigns jobs to computing resources based on its computation capabilities. First, RALBA calculates the computation share of each resource, then the jobs are allocated based on these computation shares. RALBA consists of two sub-schedulers, i.e., Fill and Spill scheduler. The fill scheduler maps jobs to resources based on their computation share. It selects a resource with higher computational capacity and assigns a large job. The fill scheduler continues this allocation until the smaller job’s length exceeds the maximum computation share of available resource. Once the fill scheduler has completed its task, the spill scheduler emerges. It assigns jobs to resources based on an earlier finish time (EFT) metric. The EFT metric considers the completion time of the jobs assigned to the resources and aims to minimize the overall job completion time. Furthermore, the computation share of a resource is determined at an earlier stage of the job scheduling as being a static algorithm. SLA-RALBA is a modified version of RALBA that considers the quality of services in the form of cost estimation and execution time in amalgamation with resource utilization in the Cloud computing environment (Hussain et al., 2019). The empirical results reveal that SLA-RALBA provides an even trade-off between execution time and cost of the services by guaranteeing a drastic improvement in resource utilization on Cloud. The dynamic variant of RALBA in the form of DRALBA is proposed by Nabi, Ibrahim & Jimenez (2021). DRALBA updates the status of the computing resource at runtime for better load balance. DRALBA has two schedulers: the Dynamic-Updater scheduler and Dynamic RALBA scheduler (Nabi, Ibrahim & Jimenez, 2021). DRALBA produces a load balanced scheduling but increases the time complexity of the algorithm.

The cloud resource broker incorporates a dynamic load-balancing to ensure a fair workload distribution across VMs in the cloud environment (Kumar & Sharma, 2020). This dynamic load-balancing algorithm optimizes resource utilization, minimizes job makespan, and enhances cloud elasticity. However, continuous resource monitoring and reallocation can affect the system’s performance and responsiveness. Shafiq et al. (2021) introduces an algorithm for efficient job scheduling in cloud computing, particularly focusing on infrastructure as a service (IaaS) models, where managing limited resources and ensuring high service delivery performance are critical. The proposed algorithm prioritizes quality of service parameters, VM priorities, and resource allocation to enhance load balancing and meet service level agreement (SLA) requirements (Shafiq et al., 2021).

The honey bee behavior-based load balancing (HBB-LB) algorithm (Ebadifard, Babamir & Barani, 2020) aims to reduce makespan, improve load distribution, and enhance system reliability by considering previous VM experience. Comparative analysis reveals that load balancing (including HBB-LB) and dynamic scheduling algorithms demonstrates reliability and load balancing improvements. However, this may still face challenges related to scalability and adaptability to highly dynamic cloud environments. Nabi & Ahmed (2021) introduces OG-RADL, a novel overall performance-based resource-aware dynamic load-balancer for deadline-constrained jobs on the cloud. It addresses the need for multi-parameter optimization in cloud job scheduling to improve overall performance. OG-RADL (Nabi & Ahmed, 2021) distributes workloads efficiently, supports deadline constraints, and exhibits superior performance compared to existing algorithms like DLBA (Mishra et al., 2017), DC-DLBA (Kumar & Sharma, 2018), Dy-MaxMin (Mao, Chen & Li, 2014), RALBA (Hussain et al., 2018), PSSELB (Mao, Chen & Li, 2014), and MODE (Yazdanbakhsh, Isfahani & Ramezanpour, 2020), as demonstrated in experimental results. While OG-RADL shows promising performance improvements, its real-world applicability and scalability in large and dynamic cloud environments need further investigation.

Traditional workflow scheduling often focuses on optimizing time or cost but lacks a comprehensive framework. Choudhary et al. (2022) presented a cloud scheduling framework offering a step-by-step solution to workflow execution, addressing energy consumption and cost. It employs power-aware dynamic scheduling algorithms, task clustering, and the partial critical path (PCP) method to organize tasks and assign sub-deadlines. Energy efficiency is enhanced using dynamic voltage and frequency scaling (DVFS), which dynamically adjusts processor voltage and frequency. Simulations on applications like Montage and CyberShake confirm that the framework effectively reduces transmission costs and energy consumption. Ma et al. (2023) focuses on optimizing VM migration by addressing three key tasks: determining migration time, selecting VMs to migrate, and choosing destination hosts. It proposes an adaptive dynamic threshold method for timing, a correlation and utilization-based strategy for VM selection, and an improved energy consumption-aware best-fit algorithm for host selection. The empirical evaluation using CloudSim and PlanetLab traces reveals that the methods achieved reductions in energy consumption, service level agreement violations, and VM migrations as compared to existing approaches. The results highlight improved service quality and energy efficiency. Murad et al. (2024b) introduces a priority-based fair scheduling (PBFS) for cloud computing to optimize resources using CPU time, arrival time, and job length. An Earliest Gap Shortest Job First (EG-SJF) technique was devised to maximize schedule utilization by effectively filling gaps. CloudSim simulations demonstrate that the PBFS algorithm outperforms LJF, FCFS, and MAX–MIN in reducing delay, makespan, and flow time. Murad et al. (2024a) focuses on enhancing Priority Rules (PR) cloud schedulers by developing a dynamic scheduling algorithm that optimizes schedule gaps. The PBFS algorithm allocates resources efficiently, while the new proposed Shortest Gap-Priority-Based Fair Scheduling (SG-PBFS) strategy manipulates schedule gaps to improve performance. Results highlight the algorithm’s effectiveness in enhancing cloud job scheduling efficiency. Table 1 presents a concrete summary of the related work by identifying the key features of scheduling algorithms in literature.

Table 1 Summary of techniques in literature.

Heuristics	Reference	Key features	
Min-Min	Elzeki, Rashad & Elsoud (2012)	Favors small size jobs, lower makespan for small size jobs, while longer wait for larger size jobs	
Max-Min	Tabak, Cambazoglu & Aykanat (2013)	Favors larger jobs, while penalize the smaller size jobs and lead to load-imbalance	
Dynamic-Max-Min	Mao, Chen & Li (2014)	Update VM status table after each iteration, while poor resource utilization and low throughput	
PSSLB	Alaei & Safi-Esfahani (2018)	High resource utilization and performance, while causing the higher cost	
PSSELB	Alaei & Safi-Esfahani (2018)	Minimize the load imbalance and SLA violation, while results in poor makespan on a higher dataset	
RALBA	Hussain et al. (2018)	Computation-aware job allocation, while causing load-imbalance as compared to dynamic algorithms	
HBB-LB	Ebadifard, Babamir & Barani (2020)	Reduce makespan, improve load distribution, and enhance system reliability, while facing challenges related to scalability and adaptability in cloud environments	
Dynamic load balancing	Kumar & Sharma (2020)	Reduce the makespan time and increase the average resource utilization ratio in cloud environment, while system’s performance degradation due to continuous resource monitoring and reallocation	
Load balancing in cloud computing environment	Shafiq et al. (2021)	Efficient task allocation in cloud computing and improve the performance, Further enhancement to provide better scheduling techniques	
DRALBA	Nabi, Ibrahim & Jimenez (2021)	Update VM status table after some interval, while increasing time complexity of algorithm	
OG-RADL	Nabi & Ahmed (2021)	Improve overall performance, while the Real-world applicability in large and dynamic cloud needs further investigation	
Power-aware dynamic scheduling	Choudhary et al. (2022)	Optimize execution using sub-deadlines; while Increasing scheduling complexity	
Energy-aware virtual machine migration and consolidation	Ma et al. (2023)	Minimize migration time and SLA violations; while possessing the limited scalability for large datasets	
Advanced Max-Min algorithm	Raeisi-Varzaneh et al. (2024)	Optimize makespan, waiting time, and resource utilization, while increasing the computational complexity	
PBFS	Murad et al. (2024b)	Maximizes schedule utilization by filling gaps effectively; while it may lead to suboptimal performance when jobs are highly variable in size	
SG-PBFS	Murad et al. (2024a)	Improving cloud job scheduling efficiency; while facing higher computational overhead in gap management	

Proposed DE-RALBA algorithm

In this section, the proposed DE-RALBA algorithm is explained by presenting its system architecture, system model, performance model and the detailed algorithm.

DE-RALBA overview

DE-RALBA is a dynamic and enhanced technique in cloud-based computing architecture. It comprises two sub-schedulers: DE-RALBA and Update-Scheduler. Figure 1 shows the DE-RALBA based Cloud computing architecture at abstract level. The performance evaluation of DE-RALBA is conducted in a simulation environment using the CloudsimPlus. The tasks or jobs on the cloud system are referred as cloudlet in CloudSimPlus simulator. The delivery of infrastructure-as-a-service (IaaS) and platform-as-a-service (PaaS) are provided by the physical and virtual layers to the cloud users. The physical layer comprises the actual computing, memory and storage resources in the form of physical machines (PMs) presenting the overall computing power of a datacenter. These physical machines are presented as PM1, PM2, PM3,…, PMN in the Fig. 1. The computing and storage services are transparently managed by the virtualization layer to provide dynamic sharing of computing and storage in the form of VMs to cloud users (Hussain et al., 2018). Cloud resource manager operates at the top of the virtualization layer and its primary responsibility is to handle the creation, termination, and migration of VMs as needed for the computing purposes. The cloud resource manager also communicates to provide the availability and computing capabilities of VMs to the DE-RALBA scheduler. To ensure efficient utilization of VMs in load-balanced manner, the proposed DE-RALBA algorithm is implemented on top of the virtualization. This algorithm provides a balance allocation of cloudlets across the available VMs in the cloud system.

Figure 1 Abstraction layer of DE-RALBA architecture.

The system accessibility layer serves as an interface for the cloud users to conveniently submit their cloudlets for processing in the cloud environment.

DE-RALBA system architecture

The proposed DE-RALBA technique is an enhanced version of the RALBA mechanism (Hussain et al., 2018). DE-RALBA system architecture is presented in Fig. 2 and the Table 2 presents the terminologies and notations used in explaining the system model of DE-RALBA. The proposed approach is dynamic scheduling which updates the VM’s status and manages the arrival time of the cloudlets at runtime.

Figure 2 Dynamic enhanced resource-aware load balancing algorithm.

Table 2 Notations used in DE-RALBA system.

Variable/Notation	Description	
cld	cld is representation of user jobs on Cloud	
vList	List of Virtual Machines in a datacenter	
cldRepository	Repository of user tasks with arrival time	
MIPS	Million Instructions Per Second (Computation Power of VM)	
MI	Million Instructions (size of cld)	
vCRatio	Computation Ratio of all the VMs	
maxPCld	Maximum possible cloudlet for a VM	
vSharej	Computation Share of VMj	
newVShare	Modified computation share of a VM	
vStatusTable	List that stores set of VM Computation Share, Ready Time, Status of VM, and list of cld associated to VM	
vStatusTablevmj	Status table of VMj in vmStatusTable	
cldStatusTable	List that stores set of cld and their status	
cldStatusTablej	Status table of cldj in cldStatusTable	
cldList	List of cld to be scheduled	
cldSList	A map <cld, VM> to store scheduled cloudlets	
cldTList	List to store temporary clds	
maxVShare	VM with maximum computation share from all VMs	
maxCld	Maximum length cld from cldList	
vEFT	Vm that give earlier finish time for maxCld	
tLenght	Sum of clds length in CldList	
isVFinishedExecution	A flag to determine if a VM completes all assigned clds	
isCldArrived	A flag to determine the arrival of new cld	
lV	VM with loaded cld from all VMs	
emptyV	VM with no clds	
minCld	A cld with minimum length	
vCT	Completion time of a VM	

The DE-RALBA comprises two sub-schedulers, i.e., DE-RALBA and Update. The DE-RALBA maps the cloudlets to VMs based on computation share and EFT of cloudlets. The computation share of each VM is determined based on the total workload submitted and the total computation power available for executing it on the Cloud. At first, VMj with maxVShare is selected and determines maxPCld for this VMj. The selected cloudlet (maxCld) is assigned to VMj and vStatusTablevmj is updated. The vmShare of VMj is modified and the assigned cld is removed from the cldList. This process continues until maxPCld returns NULL and cldList becomes empty. If maxPCld returns NULL, the scheduler checks the status of newly arrived cloudlets. If new cloudlets arrive, then the Update-Scheduler is called. If the maxPCld and iscldArrive returned false while cldList is non-empty, then the scheduler allocates the already arrived clds to VM using EFT on VM. The scheduler picks the maxCld from the already arrived cldList and determines the VMj for maxClds with EFT. The selected cld is assigned to VM and vStatusTablevmj of VMj is modified. The process continues until cldList becomes empty. The DE-RALBA Scheduler also monitors the vStatusTable of VM when the VMj completed all of its allocated clds and the remaining vStatusTablevmj.remMI of VMj is ZERO. The scheduler then tries to balance the load by shifting cld from loaded VM to emptyV (empty VM). The cldTList is getting from VMj of lV and compares the execution time of lV and emptyV; if the emptyV execution time is less than lV execution time, then cld is moved to emptyV, and ctdSList is updated. The ctdTList is iterated, and removing the minCld from the list is after one iteration. The process continues until the ctdTlist becomes empty.

The Update-Scheduler updates the vStatusTablevmj of VMj with the current time of the execution. The input of the scheduler is the list of VMStatusTable and the current time of the datacenter. The vStatusTablevmj.executedMI, vStatusTablevmj.remMI, vStatusTablevmj.readyTime of VMj are updated with respect to the time clock of the datacenter and the update vStatusTable is returned to the main scheduler.

DE-RALBA system model

The system model of cloud design delineates the performance of DE-RALBA-based job scheduling. A cloud is comprised of various VMs represented as VMS = { VM1, VM2, …, VMm}, where m is the number of VMs, and a single VM can be presented as VMj (1≤j≤m). The cloudlets are presented as CLS = { Cloudlet1, Cloudlet2, …, Cloudletn}, where n is the number of cloudlets and a single cloudlet can be shown as Cloudleti (1≤i≤n). A cloud resource manager provides the computation ratios (vmCrMap) of all VMs to DE-RALBA, where vmCrMapj can compute as:

(1) vmCrMapj=VMj.MIPS∑k=1mVMk.MIPS.

vmCrMap is used to ensure a balanced load distribution among VMs and to calculate the compute share of all VMs (VMShare). The VMSharej of VMj is expressed as:

(2) VMSharej=∑i=1nCloudleti.MI×vmCrMapj.

The scheduler assigns a cloudlet to a VM depending on the VMShare. The RPCloudletj of a given VMj may be calculated as follows:

(3) RPCloudletj={Cloudlet|Cloudlet∈CLS,Cloudlet≤VMSharej}

where, maxPCld in each scheduling decision can be computed as:

(4) maxPCld=max∀p∈RPCloudletjSize(p).

In addition, each time the scheduler makes a scheduling choice; the candidate cloudlet is deleted from the list of cloudlets. While, minCld and maxCld are computed:

(5) minCld=min∀c∈CLSSize(c)

(6) maxCld=max∀c∈CLSSize(c).

On cloudlet— VMj allocation, the computation share of VMj is modified by subtracting the cloudlet size from the current VMSharej. The VM with the biggest VMSharej is chosen by calculating the maxVShare as follows:

(7) maxVShare=max∀j∈{1,2,3,…,m}(VMSharej).

The scheduler utilizes Cld_EFTi to assign leftover cloudlets to VMs. The Cld_EFTi of a given Cloudleti is determined using the formula:

(8) Cld_EFTi=max∀j∈{1,2,3,…,m}(Cloudlet_CTij).

where Cloudlet_CTij is Cloudleti estimated completion time on VMj: The definition of the Cloudlet_CTij is:

(9) Cloudlet_CTij=(Cloudleti.MIVMj.MIPS)+VM_CTj.

The VM_CTj is the completion time of VMj for workloads already assigned before the allocation of Cloudleti and is calculated as follows:

(10) VM_CTj=∑i=1nCloudleti.MI×F[i,j]VMj.MIPS.

where F[i,j] is a Boolean variable that decides the assignment of Cloudleti to VMj, as shown:

(11) F[i,j]={1,CloudletiisassignedtoVMj0,Otherwise.

DE-RALBA performance model

Makespan is represented as Panda & Jana (2017), Hussain et al. (2018):

(12) Makespan=max∀j=1,2,3,…,m(VM_CTj).

ARUR is a metric for presenting the total cloud usage (Panda & Jana, 2017; Hussain et al., 2018) as presented in following equation. ARUR’s value stays between 0 and 1. The higher the ARUR number, the greater the resource consumption in the cloud.

(13) ARUR=∑j=1mVM_CTjmMakespan.

DE-RALBA algorithm

The DE-RALBA algorithm is elaborated with its two primary sub-schedulers i.e., DE-RALBA Scheduler (in Algorithm 1) and Update Scheduler (in Algorithm 2).

Algorithm 1 DE-RALBA.

  input: List <VMStatusTable> vStatusTable - set VMs with status table {Vm, vmShare, vCRatio, readyTime, executedMI, totalAssignedMI, lastUpdateTime, remMI}.	
  List <CloudletStatusTable> - cldStatusTable set of cloudlets with status table {cld, Vm, length, isScheduled}	
  output: cldSList map <cld, Vm> - mapping of clds to VM.	
1 maxVShare = null	
2 maxPCld = null	
3 tLength = 0	
4 maxCld = null	
5 vEFT = null	
6 newVShare = 0	
7 vCRatio = 0	
8 isCldArrived = false	
9 vList = getVmList(vStatusTable)	
10 cldList = getCloudletList(cldStatusTable)	
11 tLength = calcTotalLength(cldList)	
12 for Vm vmj : vList do	
13   vCRatio = vmj.getMips()/getTotalMips(vList)	
14   vShare = tLength * vCRatio	
15   vStatusTablevmj.vmShare = vShare;	
16 while cldList != null do	
17   maxVShare = getVm_MaxVShare(vStatusTable)	
18   maxPCld = getMaxPCloudletForVm(maxVShare)	
19   if maxPCld != null then	
20     cldSList.put(maxPCld, maxVShare)	
21     newVShare = vmStatusTablevmj.getvmShare() - maxPCld.getLength()	
22     vStatusTablevmj.vmShare = newVShare	
23     cldList.remove(maxPCld)	
24  else if isCldArrived then	
25     UpdateScheduler(currentTime, vStatusTable)	
26     isCldArrived = false	
27  else	
28     maxCld=getMaxCloudlet(cldStatusTable)	
29     vEFT=getVmWithEFT(maxCld)	
30     cldSList.put(maxCld, vEFT)	
31     vStatusTablevmj.readyTime = vmStatusTablevmj.readyTime()-getExecTime(vEFT,maxCld)	
32     cldList.remove(maxCld)	
33 Interrupt on first empty VM	
34 lV=getLoadedVm(vStatusTable)	
35 while isEmptyV(vStatusTable) do	
36        emptyV = fEmptyVm(vStatusTable)	
37        if emtpyV==null then	
38            break	
39        cldTList = lV.getCldList()	
40        while cldTList!=null do	
41         minCld = getMinimumCld(cldTList)	
42         if getExecTime(emptyV,minCld) < getExecTime(lV,minCld) then	
43          cldSList.modify(minCloudlet, emptyV)	
44          UpdateScheduler(vStatusTable)	
45         cldTList.remove(minCld)	
46 return cldSList	

Algorithm 2 Update-Scheduler.

  input: List <VMStatusTable> vStatusTable - set VMs with status table {Vm, vmShare, vCRatio, readyTime, executedMI, totalAssignedMI, lastUpdateTime, remMI}.	
  currentTime - Current Time of Simulation Clock.	
  output: vStatusTable	
1 executedMI = 0	
2 for VMStatusTable vm: vStatusTable do	
3   executedMI = vm.getComputationPower() * currentTime	
4   vm.executedMI = executedMI	
5   vm.remMI = vm.getTotalMI() – ExecutedMI	
6   vm.readyTime = currentTime - vm.getLastUpdateTime()	
7   vm.lastUpdateTime = currentTime	
8 return vStatusTable	

DE-RALBA-scheduler algorithm

The DE-RALBA is a scheduling mechanism designed to optimize cloudlet allocation across heterogeneous VMs in cloud computing environments. The algorithm addresses the critical challenges of efficient resource utilization, balanced load distribution, and reduced execution times. By leveraging the computational capacities of VMs and dynamically adapting to workload changes, DE-RALBA ensures that cloudlets are allocated in a manner that minimizes execution overhead while preventing resource underutilization or bottlenecks. The algorithm begins by analyzing the system’s computational capacity, represented in the VM Status Table (vStatusTable) and Cloudlet Status Table (cldStatusTable). The vStatusTable provides essential details about each VM, including its processing power, readiness status, and currently assigned workloads. The cldStatusTable contains information about cloudlets, such as their lengths (in terms of million instructions) and scheduling statuses. Using this data, DE-RALBA calculates the total computational workload and determines each VM’s share (vShare) based on its processing capacity ratio (vCRatio). This preliminary step ensures that each VM is assigned a workload proportional to its capabilities, promoting equitable resource distribution.

The core of the algorithm revolves around iteratively assigning cloudlets to VMs in a manner that prioritizes efficiency. VMs with the highest remaining share (maxVShare) are given precedence, and the most suitable cloudlet (maxPCld) is identified and assigned to these VMs. The algorithm dynamically updates the VMs’ remaining share to reflect the newly assigned workload and removes the cloudlet from the scheduling queue. In cases where no immediate cloudlet is suitable for the VM with the highest share, the scheduler invokes an update mechanism that incorporates newly arrived cloudlets into the system. Furthermore, if no VMs with high remaining shares are available, DE-RALBA employs an earliest finish time (EFT) heuristic. This heuristic identifies the VM that minimizes the execution time for the given cloudlet, ensuring cloudlets are completed as quickly as possible. To address imbalances caused by idle or underutilized VMs, the algorithm incorporates a robust load-balancing mechanism. If any VM remains idle while others are heavily loaded, DE-RALBA redistributes cloudlets from loaded VMs to idle ones. This redistribution is guided by a reassessment of execution times, ensuring that the overall system throughput is optimized without introducing unnecessary delays. Additionally, the algorithm handles interruptions by dynamically reevaluating the status of empty or partially loaded VMs and reallocating cloudlets to ensure consistent resource utilization.

The iterative process continues until all cloudlets have been assigned and processed. Through its dynamic, priority-based cloudlets allocation and load-balancing strategies, DE-RALBA ensures an efficient and balanced use of resources. Its ability to adapt to real-time system changes and workload variations makes it a powerful solution for modern cloud computing environments, where efficiency and scalability are paramount.

Update-scheduler algorithm

The algorithm begins by initializing a variable executedMI to 0. This variable tracks the amount of work completed by each VM. It then iterates over each entry in the vStatusTable, which contains the status of all VMs in the system. For each VM, the algorithm calculates the executedMI as the product of the VM’s computation power and the current simulation time (currentTime). This computation determines how much work the VM has processed since the last update. Following the calculation of executedMI, the algorithm updates the remMI (remaining Million Instructions) for each VM by subtracting the executedMI from the total MI of the VM (getTotalMI). This step is crucial for tracking the remaining workload that needs to be completed. The algorithm also updates the readyTime for each VM, which is the difference between the currentTime and the VM’s last update time (getLastUpdateTime). The readyTime indicates how long it has been since the last time the VM was scheduled or executed, helping to track its readiness for the next cloudlet. Finally, the lastUpdateTime of each VM is set to the current simulation time to ensure that the next update reflects the latest time point. Once all the VMs in the system have been updated, the algorithm returns the modified vStatusTable, providing the most current status of each VM, including the executed MI, remaining MI, and ready time. This updated information is essential for efficient resource allocation and cloudlet scheduling in the system.

Experimentation and computing setup

This section presents the cloud computing setup and the datasets used for the experimentation conducted in this study.

Experimental setup

The experiments are conducted on a computing machine equipped with Intel Core™ i5-4030U CPU @ 1.90 GHz 2.49 GHz and 12 GB of RAM. Table 3 describes the configuration of the computing simulation environment. The experiments are evaluated by using computing resources ranging from 16 to 50 VMs hosted on 30 host machines within a Cloud. The statistics of the VMs with its power in MIPS (Millions of instructions per second) used for the execution of various instances of the GoCJ and HCSP datasets are shown in Figs. 3 and 4.

Table 3 Configuration of the simulation environment.

Simulator/Version	CloudsimPlus version 6.3.9	
Computing power of cloud host machines:	30 Quad-core (4,000 MIPS)	
Total cloud host machines:	30	
Total VMs	Google Like dataset (GoCJ)	50 Heterogeneous VMs	
HCSP datset	16 Heterogeneous VMs	
Total cloudlets	Google Like dataset (GoCJ)	100, 200, 300, 400, 500, 600, 700, 800, 900, 1,000, 1,500, 2,000, 3,000, 4,000, 5,000, 6,000	
HCSP datset	512 cloudlets on each instance	

Figure 3 Computation power of heterogeneous VMs used for the GoCJ dataset.

Figure 4 Computation power of heterogeneous VMs in MIPS used for the HCSP dataset.

Dataset used

Two scientific benchmark datasets are used in this study. The first benchmark dataset is the expected time to compute (ETC) model of Braun (2000) in the form of HCSP instances and the second dataset is the Google-like realistic workload proposed in the form of GoCJ benchmark dataset. The GoCJ and HCSP datasets are publicly available for use on the given URLs provided in Hussain & Aleem (2018a, 2018b) and Tracy (2000), respectively.

The GoCJ dataset comprises various types of jobs, including small, medium, big, extra-large, and massive tasks. These tasks are classified and categorized within the GoCJ dataset based on their respective sizes and complexities (Hussain & Aleem, 2018b). In the HCSP dataset, a range of tasks, including small, medium, big, extra-large, and massive are included. These tasks are classified based on their size. HCSP instances are labeled using a name with pattern C-THMH, where c indicates the consistency type (c for consistent, s for semi-consistent, and i for inconsistent), TH and MH indicate the job and machine level heterogeneity, respectively (lo and hi for low and high heterogeneity, respectively). This shows that HCSP instances provide different variants of possible heterogeneity at machine’s power and job’s sizes level. Additionally, the HCSP dataset provides information about the specifications of the host machine and VMs used for computation of different instances of HCSP dataset. Figures 3 and 4 show the composition of VMs and its computation power used for GoCJ and HCSP datasets, respectively.

CloudSimPlus simulator

The CloudSimPlus simulator is used for the experimental evaluation of this work. The CloudSimPlus is an up-to-date and full-featured cloud simulation framework. It is easy to use and extend, enabling modeling and simulation of cloud computing services. Figures 5 and 6 show a few simulations of experimentation conducted using CloudSimPlus simulator.

Figure 5 Simulation results (Makespan and ARUR) in CloudSimPlus.

Figure 6 Simulation (cloudlet to VM scheduling) in CloudSimPlus.

Results and discussion

The evaluation of DE-RALBA and other schedulings namely DRALBA, PSSELB, PSSLB, and Dynamic Max-Min are conducted using the performance metrics i.e., makespan and average resource utilization ratio (ARUR). Each experiment is repeated five times and the average values obtained from these iterations are used for the analysis and comparison purposes.

Performance results: GoCJ dataset

In Fig. 7, the ARUR results of DE-RALBA, DRALBA, PSSELB, PSSLB, and Dynamic Max-Min algorithms are presented. ARUR values are depicted on a scale ranging from 0.0 to 1.0. These results are obtained by evaluating the algorithms across different numbers of GoCJ workload instances as presented.

Figure 7 ARUR results using the GoCJ dataset.

To provide a clearer representation, Fig. 8 displays the mean ARUR-based results of DE-RALBA, DRALBA, PSSELB, PSSLB, and Dynamic Max-Min algorithms in terms of percentage resource utilization. DE-RALBA achieves significantly higher resource utilization compared to the other algorithms, with values reaching 162.13%, 98.22%, 100.77%, and 534.49% higher than DRALBA, PSSELB, PSSLB, and Dynamic Max-Min, respectively. These results demonstrate the superior performance of DE-RALBA in efficiently utilizing available resources in the cloud-based environment.

Figure 8 Mean ARUR results using the GoCJ dataset.

The results shows that Dynamic-MaxMin provides the poor resource utilization because this technique favors large-sized jobs that may cause starvation for the allocation of small-sized jobs (as mentioned in the literature). DRALBA, PSSELB, and PSSLB techniques are claimed to be load-balancing algorithms in literature and provide better resource utilization as compared to Dynamic-MaxMin algorithm. The proposed DE-RALBA is a resource-ware load-balancing technique; therefore, it provides a more balanced distribution of jobs among the resources by considering their computing powers. The resources with higher power are allocated with more number of or large-sized jobs, while relatively slower resources are allocated with less number of or small-sized jobs. The ARUR-based results in Figs. 7 and 8 prove that DE-RALBA outperforms the these scheduling algorithms with significant improvements in resource utilization by ensuring the optimal balanced distribution of workload among resources.

Figure 9 illustrates the makespan-based results of DE-RALBA, DRALBA, PSSELB, PSSLB, and Dynamic Max-Min algorithms in terms of seconds, using different numbers of GoCJ instances workload.

Figure 9 Makespan results using the GoCJ dataset.

For more clarity of representations, Fig. 10 presents the average makespan in seconds achieved by DE-RALBA, DRALBA, PSSELB, PSSLB, and Dynamic Max-Min algorithms. In terms of average makespan, DE-RALBA demonstrates significant improvements compared to the other algorithms. Specifically, DE-RALBA achieves a decrease of 81.96%, 50.45%, 50.45%, and 83.02% in makespan compared to DRALBA, PSSELB, PSSLB, and Dynamic Max-Min algorithms, respectively. These results highlight the superior performance of DE-RALBA in minimizing the total execution time and improving the overall efficiency of task scheduling in the cloud environment by providing load-balanced schedule of workload.

Figure 10 Average makespan results using the GoCJ dataset.

Performance results: HCSP dataset

Figure 11 presents the ARUR-based results (on a scale of 0.0–1.0) of DE-RALBA, DRALBA, PSSELB, PSSLB, and Dynamic Max-Min algorithms for the different number of HCSP instances workload.

Figure 11 ARUR results using the HCSP dataset.

For more clarity of representations, Fig. 12 presents the mean ARUR-based results of DE-RALBA, DRALBA, PSSELB, PSSLB, and Dynamic Max-Min algorithms in terms of percentage resource utilization. DE-RALBA achieves 50.82%, 8.64%, 8.19%, and 52.35% higher resource utilization than DRALBA, PSSELB, PSSLB, and Dynamic Max-Min, respectively. HCSP instances provide different possible variants of heterogeneity in the composition of machines and jobs (discussed in the dataset section), the results in Figs. 11 and 12 prove that DE-RALBA provides significant improvements in resource utilization against these scheduling algorithms. The proposed DE-RALBA is termed as resource-aware technique because it provides consistently improved results in resource utilization for various dataset instances possessing different types of heterogeneity in the composition of machines and jobs.

Figure 12 Mean ARUR results using the HCSP dataset.

Figure 13 showcases the makespan-based results in seconds of DE-RALBA, DRALBA, PSSELB, PSSLB, and Dynamic Max-Min algorithms across different numbers of GoCJ instances workload.

Figure 13 Makespan results using the HCSP dataset.

To enhance clarity, Fig. 14 presents the average makespan in seconds derived from the results of DE-RALBA, DRALBA, PSSELB, PSSLB, and Dynamic Max-Min algorithms. Remarkably, DE-RALBA demonstrates substantial reductions in makespan compared to the other algorithms. Specifically, DE-RALBA achieves a significant decrease of 345.75%, 25.61%, 26.03%, and 227.89% in makespan compared to DRALBA, PSSELB, PSSLB, and Dynamic Max-Min algorithms, respectively. These results underscore the exceptional performance of DE-RALBA in effectively minimizing the total execution time and enhancing job scheduling efficiency within the GoCJ and HCSP instances workload scenario.

Figure 14 Average makespan results using the HCSP dataset.

Conclusion

DE-RALBA is an enhanced dynamic technique designed for cloud-based computing systems. The proposed technique, DE-RALBA, employs two schedulers (i.e., DE-RALBA and Update-scheduler) to ensure the efficient resource allocation of virtual compute resources to the user’s workload on the cloud. The CloudsimPlus simulator is used to evaluate the DE-RALBA and other competing scheduling algorithms. The empirical assesments prove that DE-RALBA has outperformed the competing algorithms in terms of makespan and resource utilization by ensuring the load-balanced workload distribution in a dynamic manner. Overall, the DE-RALBA method demonstrates its ability to effectively manage resources, minimize load imbalances, and enhance productivity in cloud computing systems through real-time allocation and intelligent task assignment.

In the proposed version of this technique, the energy consumption, fault-tolerant execution, memory and I/O requirement parameters are not considered and investigated. In the next version, these parameters can be incorporated to ensure a balanced trade-off between energy consumption and cost of the services by guaranteeing the improved resource utilization on cloud. The proposed methodology of DE-RALBA and its attained significantly improved results as compared to existing algorithms prove that DE-RALBA has the potential to be enhanced for energy optimal and fault-tolerant considerations in the next version.

Additional Information and Declarations

Competing Interests

The authors declare that they have no competing interests.

Author Contributions

Altaf Hussain conceived and designed the experiments, performed the experiments, analyzed the data, performed the computation work, prepared figures and/or tables, authored or reviewed drafts of the article, and approved the final draft.

Muhammad Aleem analyzed the data, authored or reviewed drafts of the article, and approved the final draft.

Atiq Ur Rehman performed the experiments, analyzed the data, performed the computation work, prepared figures and/or tables, and approved the final draft.

Umer Arshad analyzed the data, authored or reviewed drafts of the article, and approved the final draft.

Data Availability

The following information was supplied regarding data availability:

DE-RALBA is available at GitHub and Zenodo:

- https://github.com/au-rehman/DE-RALBA.

- Atiq ur Rehman. (2025). au-rehman/DE-RALBA: DE-RALBA Release (Version v1). Zenodo. https://doi.org/10.5281/zenodo.14796494.

The dataset is available at Mendeley Data: Hussain, Altaf; Aleem, Muhammad (2018), “GoCJ: Google Cloud Jobs Dataset”, Mendeley Data, V1, doi: 10.17632/b7bp6xhrcd.1.

The Heterogeneous Computing Scheduling Problem instances are available at: https://www.fing.edu.uy/inco/grupos/cecal/hpc/HCSP/HCSP_inst.htm.

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
