# Peer review of "DE-RALBA: dynamic enhanced resource aware load balancing algorithm for cloud computing"

_PeerJ Computer Science, doi:10.7717/peerj-cs.2739_

## Round 0.1 · original submission · Minor Revisions

The reviewers agreed on the relevance and technical soundness of the paper. Still, the presentation must be improved before publication (in particular, of Section 3), and the recent related work added. Some parts of the paper must be clarified, in particular regarding the experimental setting, and others more extensively discussed, such as the limitations of the proposed approach. The amount of work to be done amounts to a minor revision, carefully addressing the detailed comments provided by the reviewers.

·

Basic reporting

1. Having citations in the abstract section is not a good practice. The author should modify the abstract section into introduction, motivation, objective, methodology, and result wise.
2. Focusing on some recent references in the literature section is better. In the current version of the manuscript, the author has used the references up to 2021 only. So, the author should include some of the latest references in the study.
3. In Table 1, how can the author cite other references while mentioning the strengths and weaknesses of the reported literature? Check this.
4. What is the research gap found in the Literature Section that must be mentioned?

Experimental design

1. In Table 2, cld must be defined.
2. In Figure 1, what are PM1, PM2, PM3 etc. The author must explicitly mention this for better clarity.
3. In Figure 2, what is calculating the computation?
4. Lines 322 and 323 can be moved to the reference section with citations in the text within the manuscript.
5. Equation 2 assumes that all requests are equally demanding regarding their Million Instructions (MI). They don't account for variations in: Input/output (I/O) operations and Memory requirements. This may cause severe effects on tasks with high I/O or memory requirements. Justify how to avoid this.
6. From Equation 3, cloudlet size is assumed to be static and predictable. How to deal with heterogeneous cloudlets?

Validity of the findings

1. This manuscript's result analysis is insufficient to justify the developed model. Clarify the result section in a more efficient manner. The results presented in the manuscript seem to be superstitious.
2. The DE-RALBA-Scheduler Algorithm description is not clear. It is being described with respect to the algorithm written, which seems redundant. So, better explain this without mentioning the algorithmic contents. This section needs to be rewritten.

Additional comments

1. Check the incomplete reference in Line 404.

·

Basic reporting

Include visual outputs like simulation logs, graphs, or screenshots of CloudSim Plus results will greatly enhance the paper's practicality and help readers connect theory with practice.

Experimental design

Types of Outputs to Include:

Resource allocation graphs
Simulation progress details
VM scheduling and utilization results
Energy consumption and cost analysis (if applicable)

Validity of the findings

Paper is looking good but to validate the finding screenshot of cloudsimplus can be emerged.

Additional comments

Expanding the references section can solidify the paper's academic foundation.

Reviewer 3 ·

Basic reporting

• The abstract must summarize the performance evaluation results.
• The related work papers are not up to date in the domain where there are a lot of research papers are recently.
• Equations should be aligned and their fonts must be matched in the manuscript.
• Improve the graphical representation of the paper where there are more spaces between some lines.

Experimental design

• The paper could benefit from providing more detailed information about the simulation environment, parameters, and the rationale behind the choice of benchmark methods for comparison. This would enhance the reproducibility and understanding of the study.
• The results should be further analyzed, more details and further discussion of the simulation results is needed

Validity of the findings

• The conclusions section should conclude that you have achieved from the study, contributions of the study to academics and practices. In addition, list the advantages and disadvantages of the proposed solution, as well as indicate the limitations of work. Further, mention the recommendations of future works

Additional comments

• The list of references should be reformatted and cleaned up. Please check the some spells and typos.
• The authors should make their manuscript proofread by a native English speaker (lot of typos are avoidable using a speller).

---

## Round 0.2 · accepted · Accept

I carefully read the revision and the authors' response to reviewers. The authors addressed all reviewers' comments and improved the overall presentation of the paper. The paper is ready for publication.